# Molecular Basis of Müllerian Agenesis Causing Congenital Uterine Factor Infertility—A Systematic Review

**DOI:** 10.3390/ijms25010120

**Published:** 2023-12-21

**Authors:** Rajani Dube, Subhranshu Sekhar Kar, Malay Jhancy, Biji Thomas George

**Affiliations:** 1Department of Obstetrics and Gynaecology, RAK College of Medical Sciences, RAK Medical & Health Sciences University, Ras al Khaimah P.O. Box 11172, United Arab Emirates; 2Department of Paediatrics and Neonatology, RAK College of Medical Sciences, RAK Medical & Health Sciences University, Ras al Khaimah P.O. Box 11172, United Arab Emirates; subhranshu.kar@rakmhsu.ac.ae (S.S.K.); jhancy@rakmhsu.ac.ae (M.J.); 3Department of General Surgery, RAK College of Medical Sciences, RAK Medical & Health Sciences University, Ras al Khaimah P.O. Box 11172, United Arab Emirates; biji@rakmhsu.ac.ae

**Keywords:** Müllerian agenesis (MA), uterine aplasia, Mayer–Rokitansky–Küster–Hauser (MRKH) syndrome, uterine agenesis, molecular, genetics

## Abstract

Infertility affects around 1 in 5 couples in the world. Congenital absence of the uterus results in absolute infertility in females. Müllerian agenesis is the nondevelopment of the uterus. Mayer–Rokitansky–Küster–Hauser (MRKH) syndrome is a condition of uterovaginal agenesis in the presence of normal ovaries and the 46 XX Karyotype. With advancements in reproductive techniques, women with MA having biological offspring is possible. The exact etiology of MA is unknown, although several genes and mechanisms affect the development of Müllerian ducts. Through this systematic review of the available literature, we searched for the genetic basis of MA. The aims included identification of the genes, chromosomal locations, changes responsible for MA, and fertility options, in order to offer proper management and counseling to these women with MA. A total of 85 studies were identified through searches. Most of the studies identified multiple genes at various locations, although the commonest involved chromosomes 1, 17, and 22. There is also conflicting evidence of the involvement of various candidate genes in the studies. The etiology of MA seems to be multifactorial and complex, involving multiple genes and mechanisms including various mutations and mosaicism.

## 1. Introduction

Infertility is defined by the World Health Organization (WHO) as a disease of the male or female reproductive system resulting in failure to achieve a pregnancy after 12 months or more of regular unprotected sexual intercourse [1]. It can be “Primary”, denoting those who have never become pregnant, or “Secondary”, depicting those with the inability to conceive after at least one previous pregnancy [2]. Infertility affects millions of people worldwide [3]. The prevalence of infertility can vary throughout the world, but generally affects around one in five couples [4]. Infertility can be caused by different factors in males, females, and can be combined or even unexplained [3]. Common causes in females are diseases of the ovaries, fallopian tubes, uterus, endocrinal, genital tract dysbiosis or combined, and differ from country to country [5]. Similarly, the cause can commonly be obstruction of the tract, testicular failure of spermatogenesis, poor sperm quality, or endocrinal in males [3]. Uterine factor infertility (UFI) is defined as an absent uterus (absolute UFI) or as a nonfunctional uterus (non-absolute UFI) [6]. Absolute UFI can be due to congenital absence of the uterus or due to a hysterectomy later [6]. UFI can affect about 1 in 500 women of reproductive age or up to 5% of females, although the exact data are unknown [7,8].

Embryologically, the uterus, fallopian tubes, and upper part of the vagina develop from Müllerian ducts (MD) (or the paramesonephric duct). The unfused cranial end of MDs remains separated and forms the fallopian tubes. The further cranial end of MDs from both sides fuse vertically to form the uterine body and cervix, and the caudal part of MDs fuse to form the upper portion of the vagina. The vagina then canalizes and fuses with the embryonic cloaca to complete the vaginal canal [9,10]. Ovarian development in utero, on the other hand, is a complex process. Four components of the ovaries, namely the surface epithelium, ovarian stroma, primordial germ cells, and sex cords, develop from the coelomic epithelium, sub-coelomic mesoderm, yolk sac endoderm, and invagination of cortical coelomic epithelium, respectively [11]. Favorable conditions for optimal development of the female genital system are the presence of both functional X chromosomes (46 XX) and the absence of the SRY gene [9]. Various agents can influence embryogenesis to cause structural abnormalities. While drugs like diethylstilbestrol (DES), ionizing radiations, and certain infections are known to be teratogenic, others may or may not have a role [12,13,14].

Müllerian agenesis (MA), (Müllerian aplasia, complete uterine aplasia) or Mayer–Rokitansky–Küster–Hauser (MRKH) syndrome, is a rare disorder with an incidence of 1 per 4500–5000 females [15]. Although not identical, MRKH and MA are interchangeably used in the literature. MRKH is further divided into two types. When there is an isolated MA, with normal ovaries and without involvement of other organ systems, it is called type 1 MRKH. MRKH type 2 includes an absent uterus along with abnormalities in the tubes, ovaries, and urinary system. Type 2 also includes a severe form called MURCS (Müllerian duct aplasia, unilateral renal agenesis, and cervicothoracic somite anomalies) [15,16]. Some consider MURCS a separate class and classify MRKH into typical (type 1), atypical (type 2), and MURCS [17]. The exact cause of MA is largely unknown due to the heterogeneity in the published literature. The disorder was long considered to be sporadic [15]. As interest in MRKH grew, there were many reported cases of familial occurrence. Hence, there emerged a subset of patients wherein an autosomal dominant inheritance with incomplete penetrance and variable expressivity was a definite probability [18,19]. Intrinsically, it is not possible to study maternal inheritance of cases with MA because of the nature of MA, which produces absolute infertility. However, it is possible now due to advances in fertility treatment which mean that having a biological child is highly possible [20]. Thus, there occurred a series of studies to search for candidate genes and specific genetic bases of MA. While earlier studies used microarrays or small gene panels/Sanger sequencing [21,22], the scenario has changed with the use of massively parallel sequencing, including whole-exome sequencing. Newer methods have opened wider opportunities for the search for genetic causes of MA [23]. Furthermore, the studies in discordant monozygotic twins with only one twin having MA support the role of environmental factors affecting the expressivity of the genetic abnormalities [24,25,26,27,28,29,30,31,32]. Hence, through this research, we aim to explore the genetic and molecular basis of MA. This will also help in counseling couples seeking newer treatment options to achieve parenthood of biologically related offspring.

## 2. Methods

### 2.1. Search Strategy

A systematic search of the electronic databases Pubmed, Scopus, Web of Science, Embase, and Google Scholar was carried out. Medical subject handling terms (MeSH) and free-text term keywords like Mayer–Rokitansky–Küster–Hauser syndrome, Müllerian agenesis, uterine aplasia, and uterine agenesis were used in combination with gene, genome, genetic or molecular to search for data in January 2023. Thereafter, manual updates were made on a weekly basis until 10 August 2023. There was no starting date for the search. The references of relevant studies were also hand-searched if they did not belong to these databases.

### 2.2. Eligibility Criteria

#### 2.2.1. Inclusion Criteria

Studied were included if they fulfilled all the following criteria:English language articles or where English translation is available;Full-text articles reporting on human genes, genome, genetics, or molecular bases;Containing information on Müllerian duct abnormality or Müllerian agenesis or Müllerian aplasia or uterine aplasia or Mayer–Rokitansky–Kuster–Hauser syndrome.

All eligible studies published before August 2023 were included for review.

#### 2.2.2. Exclusion Criteria

Exclusions consisted of animal studies, duplicated studies, review articles, non-genetic studies, articles in languages other than English where translation was not possible, and studies where full-text articles were only available upon payment. Conference abstracts, expert opinions, and critical appraisals were also excluded.

### 2.3. Study Selection

Genetic analysis can be performed with various methods. Older studies relied on conventional comparative genomic hybridization (CGH), polymerase chain reaction (PCR), or fluorescent in situ hybridization (FISH), while current analyses are carried out using newer methods [33]. Data on the methods used in each study are also extracted, through the search. As each of these methods has distinct advantages, they are mentioned in brief.

**Array–CGH (aCGH):** This assay provides higher resolution than traditional CGH, and is used as an alternative means of genome-wide screening for copy number variations (CNVs). It combines traditional CGH principles with a microarray, and thus is not dependent on actively dividing cells. An aCGH can simultaneously detect aneuploidies, deletions, duplications, and amplifications of any locus represented on an array, as well as submicroscopic chromosomal abnormalities [34].

**Whole-exon sequencing (WES):** This allows variations in the protein-coding region of any gene to be identified, rather than in only a select few genes. It is an efficient method for detecting CNVs in potential candidate genes to identify the abnormalities possibly causing disease, as most known mutations that cause disease occur in exons. However, mutations in regulatory factors coded outside exons can be missed [35].

**Whole-genome sequencing (WGS):** This is a more advanced technique based on massive-genome sequencing, and is not dependent on the availability of predefined databases for comparison and matching [36]. It can detect abnormalities in a wide spectrum of genes. However, it is expensive and requires complex analysis [37].

After a thorough search of the databases, a total of 1308 results were retrieved. All the abstracts and study titles were screened, and duplicates were removed. Furthermore, there were a total of 1226 studies excluded, as they either did not fit the inclusion criteria, were only animal studies, included only vaginal agenesis, only abnormalities other than uterine agenesis, only gonadal dysgenesis, or did not explore the genetic basis of the disease. In a manual search of references, three case reports were found and included. Finally, 85 articles were included in the analysis. The Preferred Reporting Items for Systematic Reviews and Meta-Analysis (PRISMA) are in Figure 1.

### 2.4. Data Collection

All the authors (RD, SSK, MJ, BTG) reviewed all titles independently. The potential relevance of the studies to be included for review was agreed upon by discussion on a regular basis. Selected titles and abstracts were further screened between studies to reject the overlap of cases. Full-text copies of the selected papers were obtained and the relevant data were extracted. In the case of individual case reports, if the same patient was included in more than one study with similar characteristics and findings, only the report with a larger number of patients was included. As far as possible, single case reports were cross-checked with other reports from the same location and hospital. The decision on exclusion or inclusion was decided by discussion if the time frame and characteristics of the reported cases from the same center matched. The risk of bias was not assessed due to the nature of the studies.

## 3. Results and Discussion

The studies included only cohorts, case reports, case series, or retrospective analyses of laboratory samples. The genetic analysis was carried out using various methods. Older studies relied on conventional CGH, PCR, or FISH, while most of the newer analyses were performed using newer methods like aCGH, WES, or WGS [33]. To put the completeness of genetic analysis into context, the methods used in individual studies are considered in this review.

### 3.1. Genetic Basis of MA

A universally agreeable gene is yet to be found in the available evidence. There are elaborate investigations into candidate genes associated with MA. Out of the proposed genes, one or more were implicated in specific cohorts, but none were found in all. It is rational to propose that the genes or regulators of genes essentially involved in Müllerian duct development are most likely to be involved in MA [38]. The WNT signaling pathway genes (WNT4, WNT9B), the HOX family genes, LX1, HNF1B, and a few other candidate genes have been implicated by Mikhael et al. through WES, which was then confirmed by Sanger sequencing [39]. The copy number variants (CNVs) at different locations in chromosomes 1, 16, 17, and 22 were identified by this study [39] [Table 1]. A glossary of gene names is available as Appendix A.

To overcome the drawbacks of WES, WGS was used recently by Pan et al. In addition, to further strengthen the prior evidence on specific gene involvements, this study identified five de novo variants in nine patients with MA [41]. There are also certain case reports of the involvement of CFTR, β-catenin, and te HOXA10 gene in women with complete uterine aplasia. However, it was concluded that it is unlikely to be the causative factor [92,93,94]. In a recent study by Ragitha et al. using PCR sequencing of coding exons of the WNT4 gene in 32 women with MA and gonadal dysgenesis, single-nucleotide variations, nucleotide substitution in intronic regions not affecting the normal splicing mechanism, and synonymous polymorphism (c.861C > T; p.G287G, rs544988174) were reported. Hence, any indication of WNT4 involvement in MA was not found [76].

There are a few studies exploring the inheritance of MA in families. It was not conclusively found to have a particular inheritance pattern, although an autosomal dominant trait was suggested in a few and refuted in others [15,70,80,95,96]. The challenges were incomplete family tree availability or of a particular genetic basis. Analyzing the specific genetic composition of monozygotic twins is very helpful in the identification of genetic contributions to the pathophysiology of diseases. When a single embryo divides into two after fertilization, the resultant pregnancy is called a monozygotic twin pregnancy. As they have developed from a single embryo, it is assumed that they share identical genetic composition. Studying the genetic pattern of monozygotic twins where only one has a condition gives us an insight into additional causative factors responsible for occurrences like de novo dominant mutations, or somatic mutations in the specific tissue. In a recent study on five pairs of monozygotic twins discordant for MA, the uterine tissue remnants and blood were studied using WGS [53]. They reported a mosaic variant in ACTR3B. This variant was absent in the blood of the normal twin, had low and high allele frequency in the blood, and affected tissue of the twin with MA, respectively [53]. Few of the studies elaborated on transcriptome analysis of endometrial samples from the rudimentary tissues. In a previous study of 35 sporadic patients with MRKH, perturbations in endometrial transcriptomes were described [96] This study in uterine remnants using RNA sequencing demonstrated a large number of upregulated (1236 in MRKH type 1 and 801 in MRKH type 2) and downregulated (670 in MRKH type 1 and 373 in MRKH type 2) genes associated with MRKHS [97]. It was also found that genes encoding for estrogen receptor 1 were perturbed in a few other studies [25,83,84]. Analysis of endometrial tissue in monozygotic twins also showed similar perturbations in a recent study by Buchert et al. [53].

In a recent study by Brakta et al. 2023, genetic analysis using optical genome mapping in 87 women with MRKH and available parents revealed 14 structural variants in 17/87 (19.5%). These included deletions (*n* = 7), duplications (*n* = 3), one new translocation t(7;14)(q32;q32) (*n* = 5), a previously identified translocation-t(3;16)(p22.3;p13.3), and aneuploidies (*n* = 2). They also reported mosaicism in three cases for trisomy 12, a 7;14 translocation, and 45,X (75%)/46,XX (25%). It was concluded that the exact mechanism for MA may be mosaicisms [98]. In another study by Brendan et al. in eight individuals with MRKH, WES was used for analysis. The study reported multiple damaging and potentially damaging changes in more than one woman involving chromosomes 1, 3, 4, 7, 8, 11, 12, 20, and 22 [43]. Furthermore, few of the studies have reported no abnormality in the homeobox gene, the PAX2, WNT4, GALT, AMH, and AMHR genes, copy number changes, and AMH promotor sequence variations [17,99,100,101,102,103,104,105,106,107,108,109]. Another interesting study has reported a testis-specific protein 1-Y-linked (TSPY) gene in two women out of six with MRKHS and the 46XX karyotype [110]. Similarly, the level of galactose-1-phosphate uridyl transferase (GALT) was found to be lower in erythrocytes of women with vaginal agenesis, and two variants of the GALT gene were detected in another study [111]. This suggests that multiple genes and multiple mechanisms may be involved in the pathogenesis of MA.

### 3.2. Mechanisms of Genetic Changes

The development of female genital organs is a complex process and is influenced by the interplay of genetic, hormonal, and environmental factors. To understand the basis of genetic abnormalities resulting in MA, a brief overview of the development of MDs is essential.

During embryonic development, in both male and female fetuses, the gonadal ridges have the capacity to form either the testis or ovary until 6 weeks after conception. In the gonadal ridges, the supporting-cell lineage derived from the multipotent somatic progenitor cells is programmed to include pre-granulosa (WNT4, RSPO1, FST, and CTNNB1) genes in XX fetuses [112,113].

Müllerian (paramesonephric) ducts that give rise to most of the female reproductive tract arise around 5–6 weeks of gestation as a cleft lined by the coelomic epithelium in the urogenital ridge. Further development occurs in phases [114,115]. At first, there is a thickening of the coelomic epithelium along with expression of LHX1, and anti-Müllerian hormone receptor type II (AMHR2) [114,116,117,118]. DACH1 and DACH2 are transcriptional co-factors that act by regulating the expression of LHX1 and WNT7A and are required for the formation of MD [119,120]. In the next phase, these primordial Müllerian cells invaginate from the coelomic epithelium to reach the Wolffian duct. WNT4 expression in the mesonephric mesenchyme is essential for the Müllerian duct progenitor cells to begin invagination [116,121]. The last is the elongation phase, which begins when the invaginating tip of the Müllerian duct contacts the Wolffian duct. There is then proliferation and caudal migration of cells. There is continued elongation of MD, which eventually fuses centrally close to the urogenital sinus. MD then establishes apico-basal characteristics and develops into an epithelial tube that gives rise to the endometrium, and the surrounding mesenchyme differentiates into the myometrium of the uterus and Fallopian tubes [114,118]. The Wolffian duct plays an important role in the growth of MD, by supplying WNT9B secretion [122]. LIM1 or PAX2 are transcription factors contributing to MD growth.

SOX9 has an important role in the regression of the MDs. In male fetuses, the *SRY* (sex-determining region on the Y) gene encodes the transcription factor SOX9, which plays a vital role in gonadal differentiation. Upregulation of the expression of SOX9 in normal male development causes the development of Wolffian duct and degeneration of MDs upregulating the expression of anti-Müllerian hormone (AMH), and results in the downregulation of WNT4 expression [123,124,125,126,127,128]. Other members of the SOX family and various other factors act through upregulation or downregulation of SOX9 to control MD development. SOX3 can induce SOX9 expression, and SOX8 and SOX10 upregulate SOX9 expression [129]. Similarly, *Foxl2* downregulates SOX9, and targeted disruption of *Foxl2* leads to SOX9 upregulation in the XX gonad [130]. Prostaglandin D2 also upregulates SOX9 in the absence of SRY [131].

The role of WNT4 is crucial in the development of the internal genital tract. WNT4 is a secreted protein that functions as a paracrine factor to regulate several developmental mechanisms including the uterus, cervix, and fallopian tubes. In fetuses with XX chromosomes, the absence of SRY releases WNT4 expression, which stabilizes β-catenin and silences SOX9 [132]. β-catenin is responsible for oviduct coiling [133,134]. Many growth factors, such as LIM1, EMX2, HOXA13, PAX2 and 8, and VANGL2 are also essential for the development of reproductive organs. RSPO1 is expressed in the undifferentiated gonadal ridge of XY and XX embryos and increases in the XX gonads in the absence of SRY. RSPO1 binds to G protein-coupled receptors, stimulates the expression of WNT4, and cooperates with it to increase cytoplasmic β-catenin. The increase inWNT4/β-catenin counteracts SOX9, thus leading to the ovarian pathway [135].

There are various genetic mechanisms that are potentially involved in causing a disorder, which can occur in isolation or in combinations to result in a condition. Human genomes are dynamic entities constantly influenced by alterations. The cumulative effects of small-scale sequence alterations (caused by mutation) and larger-scale rearrangements can bring about changes in the genome over a period of time [136]. The genome contains coding regions and non-coding regions. The coding regions of the DNA are directly involved in the formation of proteins, and noncoding regions may or may not be involved in the regulation of gene expression. Regulation of gene expression occurs thanks to long non-coding RNAs and epigenetic, transcriptional, post-transcriptional, translational, and protein location effects [137].

The genes that encode for an important protein related to a particular disease or a physical attribute are called candidate genes [138]. When there are chromosomal deletions, especially of the area encoding for a specific gene, the genetic functions can be completely lost, and resulting phenotypes can be severe [139,140]. Candidate genes for MA are thus thought to be related to the development of the Müllerian or Wolffian duct, or related to regulatory factors like anti-Müllerian hormone (AMH), estrogen, and estrogen receptor. The candidate genes identified by different studies are WNT4, LHX1, HNF1B, and HOXA10 [61,65,70,75,77,116,141,142]. Other genes with possible causative roles not yet substantiated adequately are HNRNPCL1, ITIH5, LRP10, MMRP14, OR2T2, OR4M2, PAX8, PDE11A, RBM8A, SHOX, TBX6, WNT9B, and ZNF816 [Table 1].

A careful balance of levels of different proteins significantly influences the embryonic developmental processes. Duplications of genes encoding these proteins can result in extra gene copies, and the resulting alteration of gene dosing can lead to developmental defects [140]. A proposed mechanism for MA is an overdose of AMH. Duplications were found in many reported studies in multiple locations involving chromosomes 1,2,3,6,7,10,12,16,18,22 and X chromosomes [Table 2]. However, the exact mechanisms of how these changes are related to the causation of MA are yet to be verified.

Mutations involve changes in the nucleotide sequence of a short region of a genome. The number of mutations occurring is usually minimized by the inherent DNA-repair enzymes in the cell, and mutations persist only when the cellular DNA-repair mechanisms fail. The mutations on coding regions are of various types. A lot of mutations are point mutations, where one nucleotide is replaced with the other, or it can involve the insertion or deletion of one or a few nucleotides [136]. Insertions of small numbers of extra nucleotides in the polynucleotide being synthesized, or failure of some nucleotides in the frame being copied, can alter the entire sequence/codon down the frame. Such proteins are usually markedly different from the original proteins. This is called a frame-shift mutation. Silent mutations (also called synchronous) are said to have occurred when changes in the nucleotide sequence have no effect on the functioning of the genome and they do not change the encoded amino acid [165]. Missense mutations are changes that alter a codon to another one, meaning the resultant amino acid is a completely different one. The effects of missense mutations are difficult to predict. If the resultant amino acid is similar to the original one or the change is in a non-critical amino acid, the functions are retained. However, the mutant protein may have a completely different function if the change affects a critical amino acid or the new amino acid is not similar to the original. On the other hand, if the mutation results in the stopping of the translation of the mRNA prematurely because of a stop codon, this is called a nonsense mutation. Nonsense mutation results in a shortened protein, which can be non-functional and this effect depends on how much of the polypeptide is lost [165]. There are various types of mutations documented in the literature, including point mutations, frameshift mutations, and missense mutations in patients with MA [Table 2]. While some of these changes involve candidate genes directly, the significance of others is unclear. It is also noteworthy that DNA methylation levels act as a regulator of gene expression. The presence of DNA methylation, in general, prevents transcriptional activation of genes at a specific cell type [166,167]. A search for altered imprinting markers at the 11p15 imprinting control region 1 (ICR1) in 100 patients with MRKHS failed to detect any defects at that locus [168]. However, it did not rule out the possibility of imprinting alterations at other locations in the etiology of MRKHS.

In a chromosomal translocation, genetic material is exchanged between two chromosomes. Translocations were detected in five different individuals involving chromosomes 8,13, 7,14, and chromosomes 3,16 [40,98,162,163]. These were one individual with t(8;13)(q22.1;q32.1), two with t(8;13)(q12;q14), and one with t(3;16)(p22.3;p13.3). In the latest case report of these by Williams et al., about ten potential genes were identified, and four of significance were further substantiated [66]. As none of the family members in the cases had similar translocations, these were regarded as sporadic occurrences.

### 3.3. Fertility Options in Women with Müllerian Agenesis

MA means absolute infertility in females, and the diagnosis can be associated with considerable psychological trauma [20]. Hence, the management requires a multidisciplinary approach including gynecologists, fertility specialists, psychologists, clinical nurse specialists, support groups, and counselors [169]. Options for parenthood in these women include surrogacy and uterine transplantation if parents seek a biologically related baby. Adoption is an option if a biologically related baby is not desired in the first place or the options have failed, are unacceptable, or contraindicated [20].

Gestational surrogacy is a popular option in women with AUFI, especially with functioning ovaries, as in MA. The oocytes of the women with MA are collected and fused with the partner’s spermatozoa through in-vitro fertilization. The resulting embryo is then transferred to the uterus of the gestational surrogate. The couple then legally adopts the child from the mother after birth. Legal parenthood of a biologically related child is thus achieved [170]. The woman carrying the pregnancy is called a gestational surrogate. The surrogate can be a close relative of the couple (altruistic) or unrelated, which would be a commercial surrogate. A study by Petrozza et al. did not find evidence of inheritance or congenital anomalies in babies born through surrogacy [96]. However, there are legal implications for this method, which can vary in different countries. Surrogacy as such or specific surrogacy may not be legally permitted in specific countries due to sociocultural issues [20,171,172]. Countries can have different policies regarding commercial or altruistic surrogacy [173,174]. In the United Kingdom, surrogacy is permitted, but surrogacy agreements are not legally enforceable. The surrogate remains the child’s legal mother from birth, up until the parenthood is legally transferred to the intended parents. This is generally carried out after 6 weeks of birth. In case of disputes, the surrogacy arrangement is not legally enforceable [20].

Uterine transplant (UT) involves transplantation of the uterus with the cervix, ligamentous supports, and blood vessels. This results in restoring the natural anatomy. Successful pregnancy and childbirth after a UT ensure conditions akin to natural biological parenthood, which is acceptable both legally and socially. The first live birth following UT was reported in Sweden in 2014 [175]. Since then, numerous case reports and series have been published. A recent review has reported 18 live births out of 45 cases of UT [176]. Typically, the whole process involves procuring a uterus from a live or deceased donor, transplantation, immunosuppression, achieving pregnancy by embryo transfer, and delivery by cesarean section followed by hysterectomy. While live organ donation involves additional risks to the donor due to operative procedures, the clinical outcomes are speculated to be better. Immunosuppression after UT is carried out with a non-teratogenic immunosuppressive regimen and monitored for graft rejection by cervical biopsies [7,177,178]. Embryo transfer is performed using a single euploid blastocyst after 6–12 months [176]. The ensuing pregnancy is then monitored following a protocol for high-risk pregnancy care, and delivered by cesarean section at 37 weeks of gestation, or sooner if indicated. Vaginal delivery is generally not advocated due to concerns about the structural integrity of the graft and sufficiency of vascular anastomoses during contractions. Depending on the reproductive expectations, further embryo transfers can be performed. The uterus is then removed and immunosuppression is stopped to prevent further complications [179]. Although the reproductive scenario in women with MA looks encouraging with UT, it involves ethical issues encompassing both assisted reproduction and organ transplantation [180,181,182].

Adoption is an option for parenthood in couples with MA. It is a legal proceeding that creates a parent–child relationship between persons not related by blood. Adoption laws vary from country to country, and adoption is generally a long process. While it can be legally and socially acceptable, the child is not biologically related to the parents.

## 4. Conclusions

MA results in absolute uterine factor infertility. The genetic basis of MA is yet unclear and etiology is mostly multifactorial. Although many candidate genes have been identified, more studies are required to substantiate the evidence. With the advancing options for parenthood of biologically related offspring, further studies will be possible to identify candidate genes, accurate mechanisms of MA, and inheritance of this condition.

## Figures and Tables

**Figure 1 ijms-25-00120-f001:**
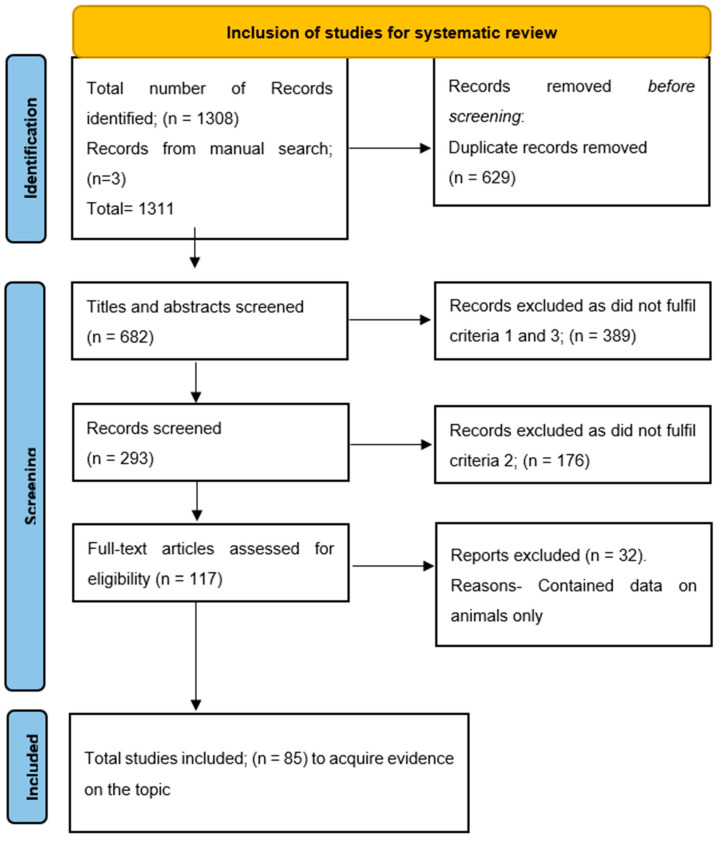
Prisma flow diagram for study inclusions.

**Table 1 ijms-25-00120-t001:** Genes suspected to be involved in MRKH.

Genes and Location in Chromosome	Reference	Method Used	Associations
MEFV and IL-32-16p13.3 CMTM7-3p22.3 CCR4 3p22.3	[40]	CGH and RT-qPCR	IL32 and MEFV gene mutations associated with Mediterranean fever. MRKHS
BAZ2B and SLC4A10-2q24.2 KLHL18-3p21.31 PIK3CD-1p36.22 TNK2-3q29	[41]	WGS	MRKHS
LAMC1-1q25.3 RARA-17q21.2 HOXA10-7p15.2 PAX2-10q24–25 MMP14 and LRP10-14q11.2	[39]	WES, confirmed by Sanger sequencing	MRKHS
IFTP57, HHLA2 and MYH15-3q13.13 PLA2R1-2q23-q24 ITGB6 and RBMS1-2q24.2	[42]	SNP microarray analysis	MRKHS
LRP10-14q11.2 FRAS1-4q21.21 CC2D2A-4p15.32 KIF14-1q32.1 RSPO4-20p13 MKKS-20p12.2 NPHP3-3q22.1 DYNC2H1-11q22.3 SPECC1L-22q11 VWF-12p13.31	[43]	WES	MRKHS
TBC1D1-4p14 KMT2D-12q13.12 HOXD3-2q31-37 DLG5-10q22.3 GLI3-7p14.1 HIRA-22q11.21 GATA3-10p14 LIFR-5p13.1 CLIP1-12q24.31	[44]	Sanger sequencing	MRKHS
PRKX-Xp22.33 HOXC8-12q13.13	[45]	RT-qPCR	MRKHS, Urinary malformations, skeletal malformations, and/or hearing defects.
MUC1-1q22	[45,46]	Array analysis, RT-qPCR	MRKHS
RBM8A-1q21	[21,47,48,49,50]	Array CGH, MLPA	TAR syndrome (thrombocytopenia, absence of radius) [21,47,48,49]
WNT9B-17q21	[38,39,49,51,52,53,54]	Array CGH, WES	MA, renal abnormalities, and cervicothoracic somite dysplasia [38]
TBX6-16p11.2	[38,39,43,48,49,50,55,56,57,58,59]	Array CGH. WES	Autism spectrum disorders, neurological disorders, unaffected persons [49]
ACTR3B-Pseudogenes chromosomes 2, 4, 10, 16, 22 and Y	[53]	WES	MA
LHX1-17q12	[21,47,49,54,55,56,60,61,62,63,64,65,66,67,68]	Array CGH, gene sequencing	MA, Anomalies in the body axis formation [49], diabetes [61], learning disability [61]
TCF2/HNF1B-17q12	[21,47,55,56,60,61,62,63,64,65,66,67,69,70]	Array CGH, DNA sequencing	MA, Renal cysts [49] diabetes [49,61], learning disability [61]
TBX1-22q11	[21,47,48,55,71,72,73,74]	Array CGH, FISH, MLPA	DiGeorge syndrome, heart defect, hypocalcemia, immunodeficiency, typical facial malformations, cognitive and behavioral disorders
WNT4-1p36.12	[39,66,75,76,77]	PCR sequencing, CGH	Hyperandrogenism (Atypical MRKHS), Gonadal dysgenesis
GREB1L-18q11.1-q11.2	[45,53,78,79,80]	WES, CGH	MRKHS type 2 with kidney abnormalities, Twins discordant for MRKHS [45]
DOCK4-7q31.1	[43,63]		MRKHS
ZNF277-7q31.1	[63]		MRKHS
DACT1-14q23.1	[81]		MRKHS
DLGH1-3q29	[82]	Direct sequencing	Unilateral agenesis, pelvic kidney
OXTR-3p25.3	[83]	DNA sequence analysis	MRKHS
ESR1-6q25	[84]	DNA sequence analysis	MRKHS
WT1-11p13	[85]	PCR	
GATA4-8p23.1	[85]	PCR	
EMX2-10q26	[38,86,87]	Sequence analysis	MRKHS, other Müllerian fusion abnormalities
SHOX-Pseudoautosomal region Xp22. 3	[39,88,89]	CGH	MRKHS
PBX1-1q23.3	[90]		MRKHS
PAX8-2q14.1	[22,38,53,88]	Array CGH. WES	Mutations have been associated with thyroid dysgenesis, thyroid follicular carcinomas, and atypical follicular thyroid adenomas [91], MRKHS

CGH = Comparative genomic hybridization; MLPA = Multiplex ligation-dependent probe amplification; WES = Whole-genome exon sequencing; WGS = Whole-genome sequencing; RT-qPCR = Reverse-transcriptase quantitative polymerase chain reaction; MA = Müllerian agenesis.

**Table 2 ijms-25-00120-t002:** Mechanisms of genetic changes.

Chromosome Number	Mechanism [Reference]	Possible Gene Involved
1	1q31.1 Duplication (size 0.4 Mb) [55] 1q44 Deletion (size 0.32 Mb) [23,48] 1p36.12 mutations 1p36.21 deletion [23] Mutation p.(Glu226Gly) [77] Mutation c.1026C>T [73] Mutation p.(Arg83Cys) [143] Mutation c.35C>T p.(Leu12Pro) [144] Mutation c.483C>T [145] Mutation c.697G>A p.(Ala233Thr) [75] Mutation g.200583493A>T [43] 1q21 Deletion (size 0.4 Mb to 4.6 Mb) [21,23,48] 1q21 Duplication (size 0.26 Mb to 0.36 Mb) [47,48]	1. KIF14 [43] 2. WNT4 is responsible for sex determination and affects the invagination of coelomic epithelial cells [121]. WNT4 mutation inhibits repression of ovarian steroid enzymes and causes abnormal expression of 17α hydroxylase and hyperandrogenism [75] 3. OR4M2, ZNF816 and PDE11A [23]
2	2p14 Duplication (size-0.23 Mb) [21] 2p14 Mutation c.1315G>A, p.Ala439Thr [53] 2p23.1 Duplication (size-0.21 Mb) [55] 2p24 Deletion (size-4.6 Mb) [55] 2q11.2 Duplication (size-1.3 Mb) [55] 2q24.2 Duplication [42] 2q13 Deletion (size-0.12 Mb) [21]	1. The PAX8 gene encodes a homeodomain signaling molecule, strongly expressed in the MD [146]. 2. Duplication at 2q24.2 of proband MRKHS involved PLA2R1, ITGB6 and RBMS1
3	3p21 Duplication 0.10 Mb [60] 3p21 Mutation c.861G>A [145] 3q13 Duplication at 3q13. [42] 3q29 Deletion 0.05 Mb [60] Mutation g.132403615G>A [43]	1. WNT7A encodes secreted signaling proteins, and is involved in the development of the anterior–posterior axis in the female reproductive tract. Coded proteins are responsible for patterning during embryogenesis. Their role in MA is uncertain. 2. May involve DLGH1, OXTR, NPHP3
4	4q28 Deletion 0.11 Mb [60] 4q32 Deletion 0.34 Mb [60] 4q35.2 Deletion 1.1 Mb [54] Mutation g79204031G>A and g.15542618C>T [43]	1. FRAS1 (g79204031G>A) [43] 2. CC2D2A (g.15542618C>T) [43]
5	5p11 Deletion 0.40 Mb [62] 5q14.3 Deletion 0.40 Mb [62]	
6	6p21 Duplication 0.17 Mb [60] 6q25.1 Duplication 0.42 Mb [60] 6q25.2 Duplication 0.44 Mb [60] 6q11.1 Duplication 0.41 Mb [62]	
7	7p15.2 Hypomethylation [32] 7p14 Duplication 1.75 Mb [60] 7q31.2 Deletion 1.8 Mb [54] Mutation g111503593C>T [43]	1. HOXA5 is a transcriptional regulator of p53 and progesterone receptor (PGR). Hypomethylation leads to overexpression, which causes overexpression of PGR [147]. Ectopic HOXA5 expression at the 5′end of the cluster might prevent normal differentiation of the MD or even regression. 2. It most likely involves Abdominal B (AbdB) homeobox genes (HOX-A9, A10, A11, and A13), required for differentiation and segmental patterning of MD [148] 3. HOXA9 is expressed in the region that becomes the oviduct [149]. Ectopic expression of HOXA5 or HOXA9 inhibits MD differentiation [150]. 4. DOCK4
8	8p23.1 Hypomethylation [85] 8p23.1 Activating mutations [32]	GATA binding protein 4 promotes AMH production and regulates sex determination and differentiation [85]. Overproduction of AMH leads to MA.
10	10q24 Duplication 0.05 Mb [60]	
11	11p11.12 Deletion 0.76 Mb (45) 11p 13 Hypomethylation [85] 11p 13 Activating mutation [32] Mutation g.102985987C>T [43]	1. WT1 is a regulatory factor important for the transcription of anti-Müllerian hormone (AMH) genes. It promotes AMH expression and regulates sex determination and differentiation [85]. Activating the mutation of the gene for the AMH receptor, or the receptor, causes excessive production of AMH, leading to MRKHS [32] 2. DYNC2H1
12	12q13.13 Duplication [42] 12q23 Duplication 0.16 Mb [60] 12q24 Duplication 0.12 Mb [60] Mutation g.6085324G>A [43]	VWF
13	13q21 Deletion 0.41 Mb [54]	
14	14q32.33 Deletion 0.46 Mb [62] 14q32.33 mutation g.23345412G>A [43]	LRP10
15	15q21.1 Deletion 0.28 Mb [62] 15q26.3 Deletion 0.54 Mb [54] Deletions at 15q11.2 [23]	
16	16p13.3 Increased expression [40] 16q11.2 Duplication 0.20 Mb [55] 16p11.2 Deletion (size 0.55 Mb to 0.6 Mb) [55] Splice site mutation c.622-2A>T (g.30100162 T>A) [56] Mutation c.484G>A(rs56098093) p.Gly162Ser [56] Mutation c.815G>A p.Arg272Gln [56] Mutation c.815G>A [54]	1. Genes for IL32 and MEFV. 2. TBX6 is involved in paraxial mesoderm formation and somitogenesis in human embryos [151]. Deletion induces MRKHS due to the loss of the transcription factor.
17	17q12 Deletion (size 1.2 to 1.9 Mb) [21,42,47,48,55,60,62] 17q12 Missense mutation of LHX1 [21,61] c.790C>G p.(Arg264Gly) c. c.25dup p.(Arg9Lysfs*25) 17q21-22 mutations c.28G>T p.Ala10Ser c.205C>T, p.Arg69Trp [53] c.*158C>T 17q21-22 Five Missense mutations [54] c.472C>G p.(Gln158Glu) c.665G>A p.(Arg222His) c.722G>A p.(Arg24His) c.974G>A p.(Arg325His) c.1029C>A p.(Cys343*)	1. Involves the LHX1(LIM homeobox protein 1) gene, which is a transcription factor necessary for the formation of the Müllerian duct-derived uterine and vaginal epithelia [116]. 2. HNF1B is [Pit–Oct–Unc homeodomain-containing transcription factor that is frequently expressed in the Müllerian duct during development [152]. It positively regulates the expression of LHX1, PAX2, and WNT9B [153]. 3. Involves mutations in WNT9B, which acts upstream of another Wnt4. It is responsible for the caudal extension of the Müllerian duct and the organization of the urogenital system [122].
18	18q23 Duplication 0.20 Mb [62] 18p Deletion [154] 18q11.1-q11.2 mutation c.4665T>A, p.Tyr1555	GREB1L is a target gene in the retinoic acid signaling pathway, which is highly expressed in the developing fetal human kidney and involved in the early metanephros and genital development [155].
19	19q13.31 deletion [23]	OR2T2 [23]
20	20q13.12 Deletion Mutations g.941074G>A and g.10393439C>A [43]	1. WISP2 is significant in smooth muscle cell proliferation and migration, and is induced by estrogen in the uterus [156]. Estrogen regulates AMH expression levels [157], and overexposure to estrogen during development activates AMH promotors [32]. 2. RSPO4 3. MKKS
22	22q11 Deletion (size 0.39 Mb to 2.6 Mb) [55,61,71,73] Duplication (0.6 Mb–3.5 Mb) [47,61,158] Mutations g.24720297G>A and g.24718408G>A [43]	1. TBX1 2. SPECC1L
X	Xp11.1 Deletion 0.12 Mb [60] Xp11.3 Duplication 0.24 Mb [60] Xp22 Duplication (0.07 to 0.36) [21,45,47,48,159] Xq21.31 Deletion 1 Mb [160] Xq deletion [161] Xq22.3 Duplication 0.09 Mb [60] Xq22.3 Microdeletion at Xp22.33 [42]	1. May involve the PRKX gene, encoding for a serine/threonine kinase implicated in renal epithelium morphogenesis [45]. 2. May involve the SHOX gene, which encodes a transcription factor responsible for skeletal growth. The exact mechanism is unknown.
8,13	t(8;13) (q12;q14) translocation	Translocation causes MRKHSS with or without renal hypoplasia [162].
	t(8;13)(q22.1;q32.1) translocation	Translocation causes MRKHS with limb, breast, and urinary functional defects [163]
3,16	t(3;16) (p22.3;p13.3) translocation	Translocation causes MRKHS [40,98]
7,14	t(7;14)(q32;q32) translocation	Translocation seen in MRKHS [98]
2, 4, 10, 16, 22 and Y	c.1066G>A, p.Gly356Arg [53]	ACTR3B encodes a member of the actin-related proteins and plays a role in the organization of the actin cytoskeleton [164]. ACTR3B can have pseudogenes in more than one chromosome.

A glossary of the genes can be found in the Appendix A; MD = Müllerian duct.

## Data Availability

No new data were created or analyzed in this study. Data sharing is not applicable to this article.

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
