# Peer review of "Molecular Basis of Müllerian Agenesis Causing Congenital Uterine Factor Infertility—A Systematic Review"

_ijms, 2023, doi:10.3390/ijms25010120_

Round 1
Reviewer 1 Report
Comments and Suggestions for Authors
In this Manuscript the authors systematically review Mullerian agenesis (MA), the nondevelopment of the uterus, in an attempt to pinpoint molecular causes for this disease, mainly focusing on genetics. Overall this is a carefully done review, mostly well planned and with appropriate inclusion/exclusion criteria, although the final number is relatively low considering the starting point. Furthermore, while several genes were identified he conclusions are not strong in terms of the possible molecular mechanisms involved. This is perhaps the main issue and often makes the manuscript difficult to read. Perhaps to address this the authors have added generic sections with limited value.
One important point is that a section on uterine development should be included (even if in animal models) to help the reader make sense of the many, often contradictory, findings. This could include a figure showing uterine development highlighting at least some large gene families/pathways known to be involved (WNT, HOX, AMH, etc). I realize this is not a main point of the review, but it is important for a non-specialized reader. Some of this information is in the tables, but hard to find systematically.
For example, It would be more important than this section:
3.2. Mechanisms of genetic changes
Which is very generic, adds nothing to this particular discussion, and could be removed to focus the review.
Although language itself is fine there seem to be organizational mistakes and the text should be thoroughly revised. For example the paragraph starting line 180 is very confusing and some sentences seem like headings of sections, such as:
Study of the specific genetic composition of monozygotic twins.
There seem to have been subsections that were removed via formatting.
Minor
The first part of Results & Discussion describing the different methodologies for molecular analysis should be in the Methods section
Comments on the Quality of English Language
The phrase construction is often odd, and suggests the use of automated translators. this should be thoroughly checked.
Author Response
Reviewer-1
In this Manuscript the authors systematically review Mullerian agenesis (MA), the nondevelopment of the uterus, in an attempt to pinpoint molecular causes for this disease, mainly focusing on genetics. Overall this is a carefully done review, mostly well planned and with appropriate inclusion/exclusion criteria, although the final number is relatively low considering the starting point.
-Furthermore, while several genes were identified the conclusions are not strong in terms of the possible molecular mechanisms involved. This is perhaps the main issue and often makes the manuscript difficult to read. Perhaps to address this the authors have added generic sections with limited value. Modified
One important point is that a section on uterine development should be included (even if in animal models) to help the reader make sense of the many, often contradictory, findings. This could include a figure showing uterine development highlighting at least some large gene families/pathways known to be involved (WNT, HOX, AMH, etc). I realize this is not a main point of the review, but it is important for a non-specialized reader. Some of this information is in the tables, but hard to find systematically.
- A section on Mullerian duct development including major groups of genes is included and highlighted
-For example, It would be more important than this section: 3.2. Mechanisms of genetic changes Which is very generic, adds nothing to this particular discussion, and could be removed to focus the review.
- A portion on Mullerian duct development including major groups of genes is included at the beginning of this section, to put into context
-Although language itself is fine there seem to be organizational mistakes and the text should be thoroughly revised. For example the paragraph starting line 180 is very confusing and some sentences seem like headings of sections, such as: Study of the specific genetic composition of monozygotic twins. -There seem to have been subsections that were removed via formatting.
-It is modified and highlighted
Minor
The first part of Results & Discussion describing the different methodologies for molecular analysis should be in the Methods section
-It is modified and highlighted
Comments on the Quality of English Language
The phrase construction is often odd, and suggests the use of automated translators. this should be thoroughly checked.
-The article is extensively checked for language and grammar, and changes are done.
Reviewer 2 Report
Comments and Suggestions for Authors
Manuscript ID: ijms-2695126
Type of manuscript: Review
Title: Molecular basis of Mullerian Agenesis causing congenital uterine factor subfertility- A Systematic review
Through this systematic review of the available literature, the authors searched for the genetic basis of Mullerian Agenesis (MA). The aims included identification of the genes, chromosomal locations, changes responsible for MA, and fertility options, in order to offer proper management and counseling to these women with MA.
Comments and Suggestions for Authors:
The manuscript is a very interesting review, but requires some minor considerations.
Title:
In the title of the review, it is indicated: “Molecular basis of Mullerian Agenesis causing congenital uterine factor subfertility”. As stated in the Abstract section: "Congenital absence of the uterus results in absolute infertility in females". The title should indicate: Mullerian Agenesis causing congenital uterine factor infertility.
Introduction:
Line 61. The acronym Müllerian agenesis (MA) is described for the first time in the text here, however the expression is later used again instead of the acronym. The use of acronyms should be reviewed throughout the manuscript.
In response to your request to consider providing some additional, specific comments to the revision of Manuscript ID: ijms-2695126 Type of manuscript: Review Title: Molecular basis of Mullerian Agenesis causing congenital uterine factor subfertility- A Systematic review.
I would like to inform you that I have already indicated to the authors several comments to consider on the manuscript. The rest of the issues do not bother me personally and I consider the review very interesting to publish in IJMS, if the rest of the reviewers and the editor agree.
I'm going to answer the questions you asked me:
1. Do you consider the topic original or relevant in the field? Does it address a specific gap in the field? My answer in the review (Is the work a significant contribution to the field?) was positive 3/5. Indeed, currently this systematic review address a specific gap in the field.
2. What does it add to the subject area compared with other published material? As this is a systematic review, the material analyzed has already been published, but what it attempts is a unification of the results obtained in other studies carried out previously.
3. What specific improvements should the authors consider regarding the methodology? What further controls should be considered? The methodology is adequate.
4. Are the conclusions consistent with the evidence and arguments presented and do they address the main question posed? Yes.
5. Are the references appropriate? My answer in the review (Are there appropriate and adequate references to related and previous work?) was positive 4/5.
6. Please include any additional comments on the tables and figures. The tables and figures are adequate and systematic. I hope I've been helpful. I reiterate my decision to consider the manuscript as a very interesting review, but requires some minor considerations. I have made comments to the authors. Best regards.
Comments on the Quality of English LanguageMinor editing of English language required.
Author Response
- Through this systematic review of the available literature, the authors searched for the genetic basis of Mullerian Agenesis (MA). The aims included identification of the genes, chromosomal locations, changes responsible for MA, and fertility options, in order to offer proper management and counseling to these women with MA. Comments and Suggestions for Authors:
The manuscript is a very interesting review, but requires some minor considerations.
- Title:
In the title of the review, it is indicated: “Molecular basis of Mullerian Agenesis causing congenital uterine factor subfertility”. As stated in the Abstract section: "Congenital absence of the uterus results in absolute infertility in females". The title should indicate: Mullerian Agenesis causing congenital uterine factor infertility.
-Changed and highlighted
- Introduction:
Line 61. The acronym Müllerian agenesis (MA) is described for the first time in the text here, however the expression is later used again instead of the acronym. The use of acronyms should be reviewed throughout the manuscript.
-Changed and highlighted throughout the text and tables
In response to your request to consider providing some additional, specific comments to the revision of Manuscript ID: ijms-2695126 Type of manuscript: Review Title: Molecular basis of Mullerian Agenesis causing congenital uterine factor subfertility- A Systematic review.
I would like to inform you that I have already indicated to the authors several comments to consider on the manuscript. The rest of the issues do not bother me personally and I consider the review very interesting to publish in IJMS if the rest of the reviewers and the editor agree. I'm going to answer the questions you asked me:
- Do you consider the topic original or relevant in the field?
-Does it address a specific gap in the field? My answer in the review (Is the work a significant contribution to the field?) was positive 3/5. Indeed, currently, this systematic review addresses a specific gap in the field.
- What does it add to the subject area compared with other published material?
-As this is a systematic review, the material analyzed has already been published, but what it attempts is a unification of the results obtained in other studies carried out previously.
- What specific improvements should the authors consider regarding the methodology? What further controls should be considered?
-The methodology is adequate.
- Are the conclusions consistent with the evidence and arguments presented and do they address the main question posed? Yes.
- Are the references appropriate? My answer in the review (Are there appropriate and adequate references to related and previous work?) was positive 4/5.
- Please include any additional comments on the tables and figures.
The tables and figures are adequate and systematic. I hope I've been helpful. I reiterate my decision to consider the manuscript as a very interesting review, but requires some minor considerations. I have made comments to the authors. Best regards.
Comments on the Quality of English Language- Minor editing of English language required.
The article is checked again for language and grammar, and minor changes done.
Reviewer 3 Report
Comments and Suggestions for Authors
This is an interesting systematic review on the genetic and molecular basis of Mullerian Agenesis (MA). The study concludes that the etiology of MA is multifactorial and complex involving multiple genes, and mechanisms including various mutations and mosaicism. Methodology and analysis is valid. Results are clearly presented and conclusions are supported by the results. I believe the article contributes to the current knowledge and I support its publication.
Author Response
This is an interesting systematic review on the genetic and molecular basis of Mullerian Agenesis (MA). The study concludes that the etiology of MA is multifactorial and complex involving multiple genes, and mechanisms including various mutations and mosaicism. The methodology and analysis is valid. Results are clearly presented and conclusions are supported by the results.
I believe the article contributes to the current knowledge and I support its publication.
Thank you
Round 2
Reviewer 1 Report
Comments and Suggestions for Authors
The authors have adequately addressed concerns